# Dendrite regeneration in *C. elegans* is controlled by the RAC GTPase CED-10 and the RhoGEF TIAM-1

**Harjot Kaur Brar**[1], **Swagata Dey**[1], **Smriti Bhardwaj**[1], **Devashish Pande**[1], **Pallavi Singh**[1], **Shirshendu Dey**[2], **Anindya Ghosh-Roy**[1]*

**1** Department of Cellular & Molecular Neuroscience, National Brain Research Centre, Manesar, Haryana, India, **2** Fluorescence Microscopy Division, Bruker India Scientific Pvt. Ltd., International Trade Tower, Nehru Place, New Delhi, India

* anindya@nbrc.ac.in

**Data Availability Statement:** All relevant data are within the manuscript and its Supporting Information files.

## Abstract

Neurons are vulnerable to physical insults, which compromise the integrity of both dendrites and axons. Although several molecular pathways of axon regeneration are identified, our knowledge of dendrite regeneration is limited. To understand the mechanisms of dendrite regeneration, we used the PVD neurons in *C. elegans* with stereotyped branched dendrites. Using femtosecond laser, we severed the primary dendrites and axon of this neuron. After severing the primary dendrites near the cell body, we observed sprouting of new branches from the proximal site within 6 hours, which regrew further with time in an unstereotyped manner. This was accompanied by reconnection between the proximal and distal dendrites, and fusion among the higher-order branches as reported before. We quantified the regeneration pattern into three aspects–territory length, number of branches, and fusion phenomena. Axonal injury causes a retraction of the severed end followed by a Dual leucine zipper kinase-1 (DLK-1) dependent regrowth from the severed end. We tested the roles of the major axon regeneration signalling hubs such as DLK-1-RPM-1, cAMP elevation, *let-7* miRNA, AKT-1, Phosphatidylserine (PS) exposure/PS in dendrite regeneration. We found that neither dendrite regrowth nor fusion was affected by the axon injury pathway molecules. Surprisingly, we found that the RAC GTPase, CED-10 and its upstream GEF, TIAM-1 play a cell-autonomous role in dendrite regeneration. Additionally, the function of CED-10 in epidermal cell is critical for post-dendrotomy fusion phenomena. This work describes a novel regulatory mechanism of dendrite regeneration and provides a framework for understanding the cellular mechanism of dendrite regeneration using PVD neuron as a model system.

## Author summary

The knowledge of the repair of injured neural circuits comes from the study of the regeneration of injured axons. The information receiving neurites, namely dendrites, are also vulnerable to physical insult during stroke and trauma. However, little knowledge is available on the mechanism of dendrite regeneration since the study of Cajal. In order to get

**Funding:** The Department of Biotechnology, ministry of science and technology, DBT/Wellcome Trust India Alliance (Grant # IA/I/13/1/500874) to AGR, and DBT/Wellcome Trust India Alliance (Grant # IA/E/18/1/504331) to SD.Caenorhabditis Genetics Center (CGC) is supported by the NIH Office of Research Infrastructure Programs (P40 OD010440). The funders had no role in study design, data collection and analysis, decision to publish, or preparation of the manuscript.

**Competing interests:** The authors have declared that no competing interests exist.

insight into this process, we severed both axon and dendrites of PVD neuron in *C. elegans* using laser. By comparing the roles of axon regeneration pathways in both dendrite and axon regeneration in this neuron, we found that dendrite regeneration is independent of molecular mechanisms involving axon regrowth. We discovered that dendrite regeneration is dependent on the RAC GTPase CED-10 and GEF TIAM-1. Moreover, we found that CED-10 plays roles within both neuron and in the surrounding epithelia for mounting regeneration response to dendrite injury. This work provides mechanistic insight into the process of dendrite repair after physical injury.

## Introduction

The functional nervous system of an organism requires intact neuronal processes and synaptic connections for proper transmission of electrical signals. A deficit in the structural integrity in the cognitive areas of brain leads to manifestation of neuropathologies[1–4]. Due to their sensitivity towards excitatory and inhibitory inputs, dendrites are often the sites of neurotoxic damage leading to severe dendritic dystrophy such as formation of dendritic varicosities, loss of dendritic spines, mitochondrial swelling and dysfunction and disruption of microtubules [5–7]. One or more of these hallmarks of dendrite damage have also been observed in focal stroke or anoxic depolarization[8], mild Traumatic Brain Injury (mTBI)[9], and epilepsy[10]. Though these features may appear neuroprotective and reversible in favorable conditions, their frequent or chronic occurrence may be devastating or fatal. Unlike axonal damage and regeneration, dendrite regeneration has not been comprehensively explored.

The knowledge about neurite regeneration has been attained mostly from axonal injury models. An injury to the axons elicits a local calcium increase [11,12] that triggers elevation in Cyclic Adenosine monophosphate (cAMP) levels, activation of downstream Protein Kinase A (PKA), and mitogen-activated protein kinase kinase kinase (MAPKKK) Dual Leucine Zipper Kinase (DLK-1) [13–15]. DLK-1 initiates local microtubule remodeling [16] and activates Ets-C/EBP-1 transcription complex promoting axon regeneration [17]. The Dendritic arborization (*da*) neurons in *Drosophila* have been recently established as an efficient model for studying dendrite regeneration [18,19]. Both intrinsic and extrinsic mechanisms of neurons can regulate the efficiency of dendrite regeneration [20]. The dendrite regeneration is independent of Dual Leucine zipper Kinase (DLK) MAPK pathway [21], which is an essential factor for the initiation of axon regeneration [13]. However, other kinases like AKT, and Ror have been implicated in the process [18,22]. Also, Wnt effectors, which regulate the dendritic morphology and branching, can also regulate dendrite regeneration process [22,23]. Although some of the cytoskeleton-based mechanisms controlling the axon regrowth do not affect dendrite regeneration [24], microtubule minus-end binding protein, Patronin-1 controls both axon and dendrite regeneration [25–27]. The roles of the axon regeneration machineries have not been extensively tested for dendrite regeneration.

PVD neurons in *C. elegans*, which is responsible for proprioception and harsh touch sensation, have an elaborate dendritic branching pattern [28,29]. Laser-induced small damage to the dendrites of PVD neurons triggers a regenerative self-fusion process [30,31]. The Fusogen AFF-1 plays a crucial role in promoting fusion between the proximal and distal dendrites after injury [30]. However, the early signalling mechanisms initiating dendrite regrowth remain elusive.

In this report, by combining 2-photon laser neurosurgery and quantitative imaging, we have established both axon and dendrite injury paradigms using the PVD neurons in worms.

Using both dendrite and axon regeneration assays in the same neuron, we assessed the roles of axon regeneration pathways in dendrite regeneration. Our results showed that the dendrite regeneration involves multiple cellular processes comprising regrowth, branching, and fusion events independent of conventional axon regeneration pathways, including DLK-1/MLK-1. Our results highlight the neuronal and epidermal roles of Rac GTPase, CED-10 in the initiation of dendrite regrowth and self-fusion processes. We also showed that TIAM-1, a Rho Guanine Exchange Factor (Rho GEF) acts upstream to CED-10 for dendrite regrowth and branching.

## Results

### Primary dendrite injury in PVD neuron triggers multiple responses involving regrowth, branching, and fusion

In *C. elegans*, PVD neurons are located mediolaterally with a well-defined ventrally targeted axon and a dendritic structure spanning anterior-posterior direction with their orthogonal arbors reaching the dorsal and ventral midline (Fig 1A). These dendrites are hierarchically classified from primary to quaternary based on their order of branching [32]. Previous studies have elucidated that following injury, primary major dendrites of PVD neurons sprout neurites, which quickly fuse with their distal counterparts with the help of the fusogen, AFF-1 delivered from the epidermal cells [30]. The mechanisms that initiate regeneration process after dendrotomy are yet to be investigated. Using GFP, and mCherry::RAB-3 labelled PVD neurons (Fig 1A), we identified the axonal and primary dendritic compartments of PVD and performed dendrotomy with a modified paradigm (Fig 1A). We delayed the self-fusion process by creating a big gap between the proximal and distal parts of the primary dendrite by using two successive laser-shots at 10–15 μm apart (Red arrow, Fig 1B). In this paradigm, regrowing processes are emanated from the severed end of primary dendrite as well as from the adjacent proximal tertiary dendrites (3h post-dendrotomy). These regrowing processes subsequently branched more (6h post-dendrotomy) (Fig 1B) and eventually reconnected with the distal dendrites at 12h and 24h post-injury (green arrowheads, Fig 1B). Due to a significant overlap between proximal and distal dendrites, we classified the reconnections based on their relative depth of the counterparts in a confocal image using a "depth-coded" projection (S1G Fig). The overlaps with the proximal and distal dendrites on the same depth (same color in the depth coded image) were quantified as reconnection events as opposed to differently colored proximal and distal counterparts (S1G Fig). This allowed a correction of false reconnections from 80% to 60% (S1G, S1H and S1I Fig). The dendrite regeneration process was also accompanied by menorah-menorah fusions [30], in which the tertiary dendrites adjacent to the injury site merge with each other (red semi-transparent box, Fig 1B) to bypass the gap of injury, and degeneration of the distal part (Fig 1B, grey traces in schematics). As time progressed (48h), the regrowing dendrite expanded the territory further with an increased number of branches (Fig 1B). The longest regrowing neurite (yellow dotted trace, Fig 1B) was used to quantify the territory covered by these regenerated neurites, termed 'territory length'. The territory length increased with respect to time after dendrotomy (Fig 1C). The regenerating branches also increased in number with time after dendrotomy (Fig 1D). The other parameters, such as 'reconnection' with the distal dendrite and 'menorah-menorah fusion' also increased with time after injury (Fig 1E and 1F). As both dendrite regrowth and reconnections happened concurrently, we asked whether the reconnection or fusion process could prevent the extent of dendrite regeneration.

To delay the reconnection process further, we severed the primary dendrites using four consecutive shots, which created a bigger gap ~100μm (S1A Fig). The gap between proximal

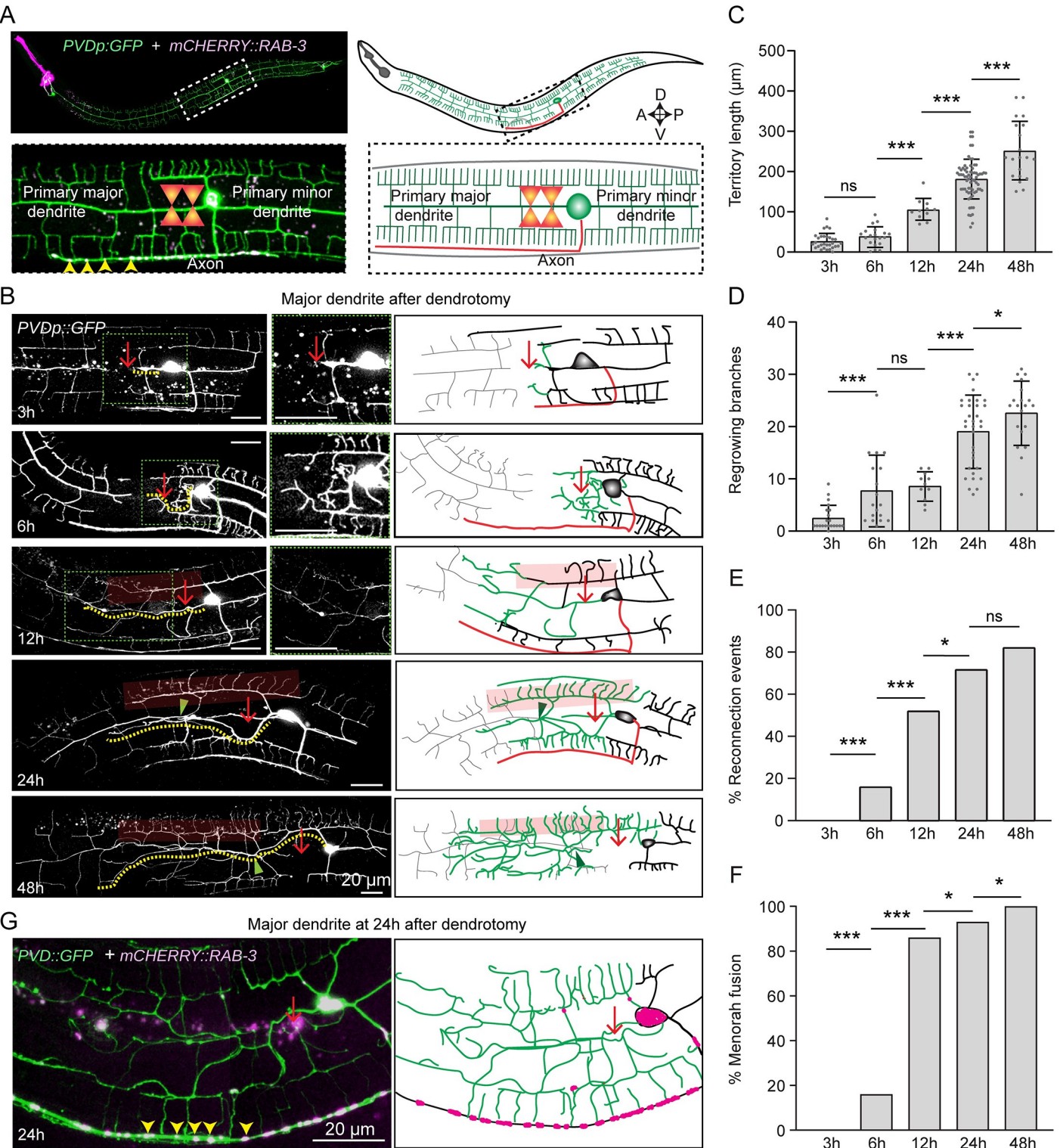

**Fig 1. Primary dendrites of PVD neurons show multiple regenerative responses following dendrotomy.** (A) Representative confocal image and illustration of dendrotomy paradigm in PVD neuron labeled with soluble GFP *wdIs52* (*pF49H12.4::GFP*) and axon marked with mCherry::RAB-3 (*kyIs445,pdes-2::mcherry::RAB-3*) with the region of interest (white dashed rectangle) magnified below. The primary dendrites and the axon are highlighted. The yellow arrowheads mark the axonal RAB-3 punctae. The dendrites and axons are illustrated in green and red color respectively. The two-shot dendrotomy cuts using a femtosecond laser is also illustrated. (B) The confocal images and schematics (right) showing the PVD neurons in the wild-type background at 3h, 6h,12h, 24h and 48h post-dendrotomy using two laser shots. At 3h after injury, the gap caused due to the two laser shots is shown with a red arrow. A magnified view of the regrowing end within the green dotted

box is shown on the right. The faded red boxes highlight menorah-menorah fusion events, and the green arrowhead represents reconnection events between the proximal and distal primary dendrites. The regenerated part and the distal remnants are shown in green and grey colors in the illustration of the regeneration events, respectively, whereas the axon is shown in red. The longest regrowing dendrite is indicated with a yellow dotted line. (C) Quantification of the longest regrowing dendrite, which is termed as 'territory length'. N = 3–10 independent replicates, n (number of regrowth events) = 20–70. (D) Number of regrowing branches in each timepoint. N = 3–7 independent replicates, n (number of regrowth events) = 14–34. (E-F) The percentage occurrence of reconnection events and the menorah-menorah fusion events. For E-F, N = 3–10 independent replicates, n (number of regrowth events) = 20–60. (G) Confocal images of dendrotomized PVD neuron at 24h post-dendrotomy, expressing GFP (*F49H12.4::GFP*) *wdIs52* and (*pdes-2::mCherry::rab-3*) *kyIs445*. The localization of RAB-3 punctae is indicated using yellow arrowheads in the confocal image and red dots in the schematics. Statistics, For C-D, one-way ANOVA with Tukey's multiple comparison test, $p < 0.05^*$, $0.01^{**}$, $0.001^{***}$. For E-F, Fisher's exact test, $p < 0.05^*$, $0.01^{**}$, $0.001^{***}$. Error bars represent SD. ns, not significant.

and distal dendritic parts due to multiple shots was significantly bigger as observed at 3-6h after injury (orange double-headed arrow, S1A Fig). Although the menorah-menorah fusion and reconnection events were significantly lower in multi-shot experiments (S1B and S1C Fig), both the territory length and branching in this experiment were comparable to the two-shot dendrotomy experiments (S1D and S1E Fig). This suggested that the reconnection or fusion processes do not influence the regenerative growth initiated upon dendrotomy in PVD neurons. Unlike axon regeneration [33], dendrite regeneration did not disrupt the axon-dendrite compartmentalization as the synaptic reporter mCherry::RAB-3 mostly remained at the ventral cord (yellow arrowheads, Fig 1G) and did not invade the regenerated neurites after dendrotomy.

Additionally, we have performed dendrotomy on the minor dendrite (S1F Fig). We observed similar regrowth response and fusion events in minor dendrites as well following dendrotomy. (S1F Fig).

Hence, both major and minor dendrites of PVD neurons can regenerate after dendrotomy irrespective of the size of the injury. The regrowing dendrites cover up the injury area in a pattern different from the original arbor. In the event of an encounter with the distal remnants, the regrowing dendrites may fuse or reconnect and integrate into the original arbor. However, this does not prevent the unfused dendritic tips from growing further. This was indicative of a molecular mechanism of dendrite regeneration underlying multiple cellular processes such as regrowth, branching, and cell fusion.

## Dendrite regeneration in PVD neurons is independent of DLK/MLK pathway

The cellular and molecular mechanisms of axon regeneration have been extensively studied using various model organisms[34]. The conserved Mitogen-activated protein kinase kinase kinase (MAPKKK) pathway involving DLK-1 is essential for the initiation of regrowth from the cut stump of axon in multiple model organisms, including mammals [13,14,35], (Fig 2A). Therefore, the initiation of dendrite regrowth might rely on the DLK-1 mediated injury response.

At 24h following dendrotomy, the primary major dendrite in *dlk-1(0)* regrew like the wild type (Fig 2B). The regenerative branching in the mutant was accompanied by reconnection of the primary dendrites (green arrowheads) and fusion between the tertiary dendrites equivalent to that of wild type (red transparent box) (Fig 2B). Since DLK-1 and MLK-1 cooperate to activate the regeneration response [36], we also tested the single mutant *mlk-1(0)* and the double mutant lacking both *dlk-1* and *mlk-1*. In *mlk-1(0)* and *dlk-1(0); mlk-1(0)*, dendrite regeneration was unaffected (Fig 2B) and the quantitative parameters like territory length, number of regrowing branches in *dlk-1(0)*, *mlk-1(0)* and *dlk-1(0);mlk-1(0)* were comparable to the wild type (Fig 2B, 2C and 2D). Similarly, the reconnection phenomena and menorah-menorah

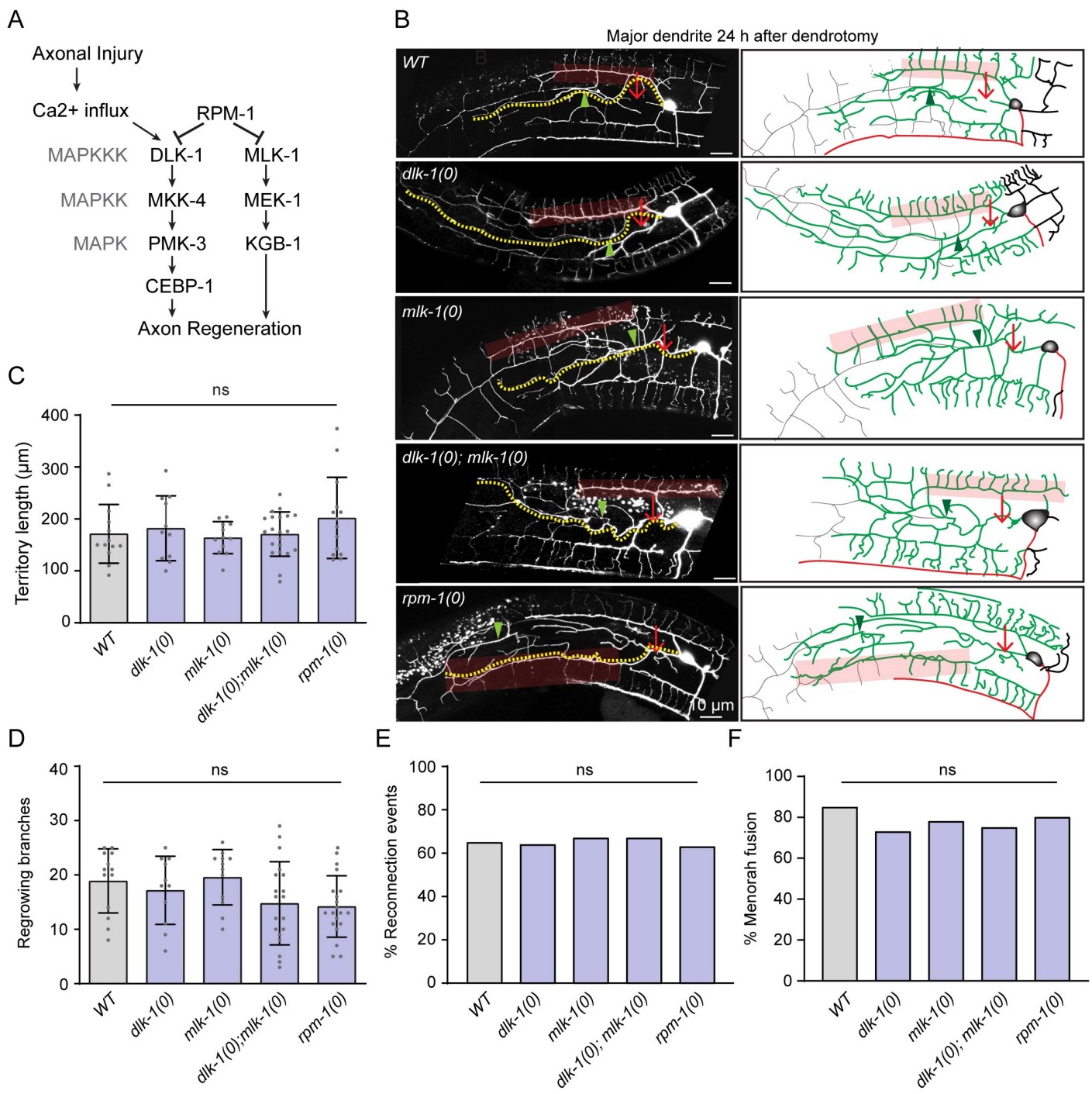

**Fig 2. The dendrite regeneration is independent of DLK/MLK pathways.** (A) Signaling pathway involving DLK-1 MAP kinase responsible for the initiation of axonal regeneration following Axotomy. (B) Confocal images of the regeneration events of primary major dendrites in the wild type, *dlk-1(0)*, *mlk-1(0)*, *dlk-1(0); mlk-1(0)* and *rpm-1(0)* at 24h post-dendrotomy. The experiment was done in the in *wdIs52* (*pF49H12.4::GFP*) reporter background. The illustrations on the right indicating site of dendrotomy (red arrow), regenerating dendrites (green color), distal part (grey color), territory length (yellow dotted lines), reconnection events (green arrowheads) and menorah-menorah fusion (faded red boxes). (C-F) Quantification of the territory length (C), total number of branches (D), the percentage of reconnection events (E), and the percentage of menorah-menorah fusion events (F) in the wildtype, *dlk-1(0)*, *mlk-1(0)*, *dlk-1(0);mlk-1(0)*, and *rpm-1(0)* at 24h post-dendrotomy. N = 3–5 independent replicates, n (number of regrowth events) = 15–20. Statistics, for C-D, One-way ANOVA with Tukey's multiple comparison test, $p<0.05^{*}$, $0.01^{**}$, $0.001^{***}$ and for E-F, Fisher's exact, $p<0.05^{*}$, $0.01^{**}$, $0.001^{***}$. Error bars represent SD. ns, not significant.

fusion events were equivalent in these mutants as compared to wild type (Fig 2E and 2F). The dendrite regeneration was also unchanged in the loss of function mutant of E3 ubiquitin ligase, RPM-1 (Fig 2), which downregulates DLK-1 and downstream kinases in the cascade during developmental growth of axon(Fig 2A) [37]. The dendrite regrowth and its ability to fuse at 24h after injury in *rpm-1(0)* was similar to the wild type (Fig 2), suggesting that *dlk-1* and *mlk-1* are neither necessary nor sufficient for the dendrite regrowth following injury in PVD neurons.

Furthermore, we checked the dendrite regeneration in the minor dendrite of *dlk-1(0);mlk-1 (0)* double mutant, which was comparable to the wild-type (S2A, S2B and S2C Fig). These observations corroborated the earlier results in *Drosophila da* neurons where dendrite regeneration was independent of the DLK-1 signalling [21]. Although *dlk-1* is expressed in PVD [38], its role in PVD neuron is unclear. Since a well-known role of E3 Ubiquitin ligase RPM-1 and downstream MAPKKK DLK-1 is to stabilize synaptic growth along with axon growth during development (S2D Fig) [37,39–41], we looked at the possible phenotype related to axon development in *rpm-1* mutant. Both the *ju23* and *ok364* alleles of *rpm-1* showed an overgrowth of axons along the ventral cord (S2E Fig). The length of the axon was significantly higher in *rpm-1* mutants (S2F Fig), and axon overgrowth phenotype was completely suppressed by loss of *dlk-1* in *rpm-1(0)* background (S2E and S2F Fig) as seen in other neurons in *C. elegans* [37] and other organisms [39,40]. This indicated that *rpm-1/dlk-1* cascade is functional in PVD neurons and strengthened our observation of unaffected dendrite regeneration in *dlk-1/mlk-1* mutants.

## Axon regeneration in PVD neurons depends on DLK-1 and MLK-1

Our finding that dendrite regeneration in PVD is independent of DLK-1 cascade raises the question of whether the axon regeneration in this neuron would require this MAP Kinase pathway. We performed axotomy at 50 μm away from the soma (Red arrow, Fig 3A) and found that the severed end retracted at 3h post-axotomy, and afterwards followed by a regrowth from the severed end (Fig 3B and 3C, green traces). The punctae of axonal reporter mCherry::RAB-3 were localized at the tip of this regrowing neurite (Fig 3B, yellow arrowheads). These punctae are often relocalized at the adjacent dendrites (Fig 3B, blue traces, orange arrowheads), suggesting the conversion of some of the adjacent tertiary dendrites into an axon. This observation was reminiscent of *Drosophila da* neurons, where the dendrites are converted to axons following a proximal axotomy [33]. The soma and the proximal part of the severed axon often emanated some ectopic processes (Fig 3B, orange traces). There was a significant extension of the axon from the severed end at 24h and 48h as compared to 3h post-axotomy (Fig 3B and 3C). Similarly, there was an increase in the conversion of adjacent dendrites to axon-like branches (Fig 3C) and the number and length of ectopic branches (Fig 3C).

We then carried out the axotomy in loss of function mutants of *dlk-1* and *mlk-1* (Fig 3D). At 24h post-axotomy, wildtype worms showed an average regrowth of 26.07±17.4 μm from the severed end which decreased significantly, due to loss of either *dlk-1* (10.17±9.93 μm) or *mlk-1* (9.04±10.59 μm) or both (8.86±11.43 μm) (Fig 3D and 3E), with negligible regrowth in nearly 50% of the mutant worms. Length of the ectopic neurites during regrowth was also reduced in the double mutant as compared to wildtype (Fig 3F). This confirmed the requirement of *dlk-1* and *mlk-1* in the PVD axon regeneration.

Thus, the axon regeneration requires DLK/MLK pathway in PVD neurons, but the dendrite regeneration is not dependent upon this signaling pathway, as seen in *Drosophila* [21]. However, dendrite regeneration might rely on other molecular pathways regulating axon regeneration.

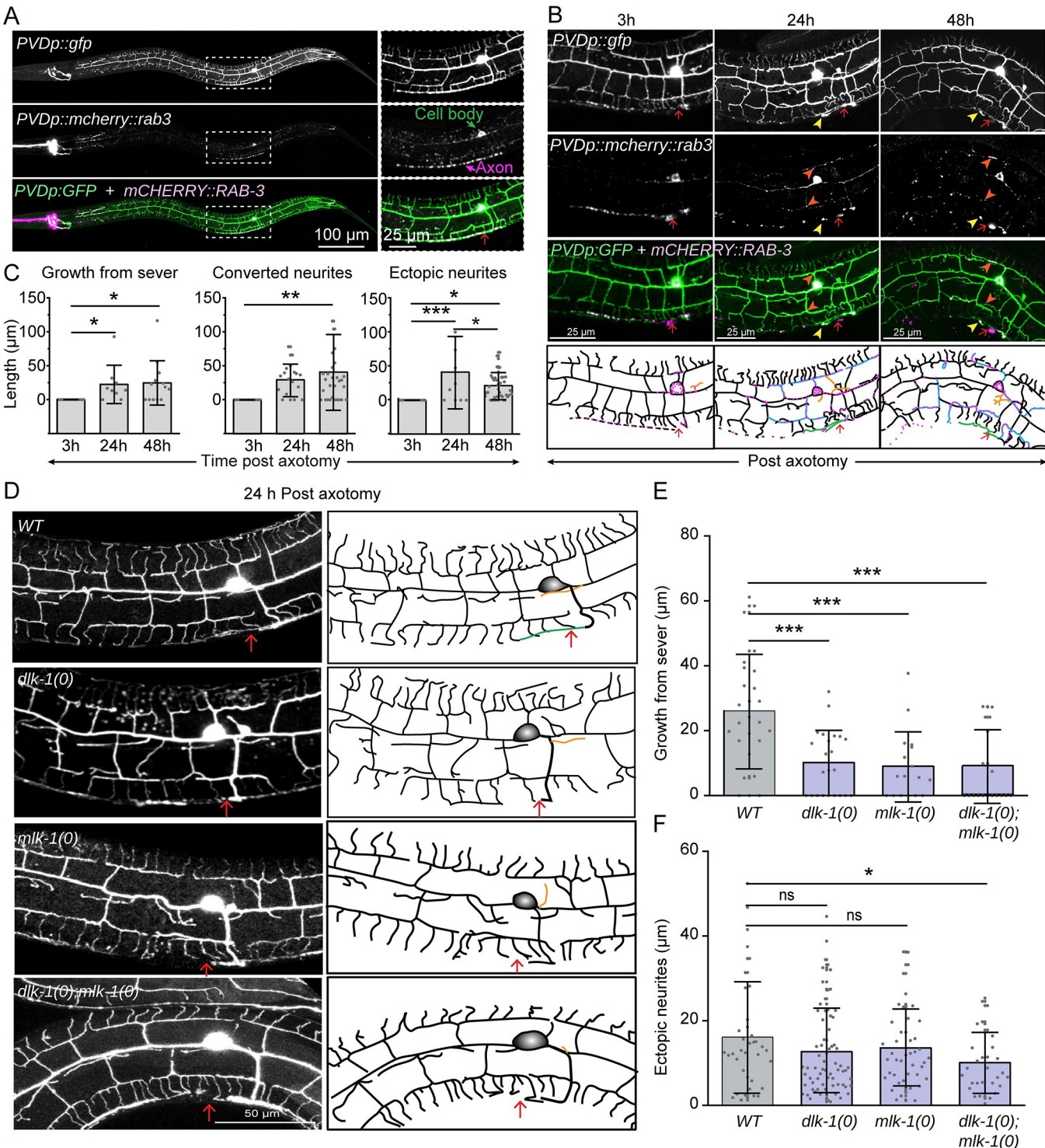

**Fig 3. Axon regeneration in PVD neuron requires the DLK-1/MLK-1 pathway.** (A) Representative images of PVD neuron labeled with *wdIs52* (*pF49H12.4::GFP*, green) and *kyIs445* (*pdes-2::mCherry::RAB-3*, magenta) with the region of interest (white dashed rectangle) magnified in the respective insets. mCherry::RAB-3 punctae localized to the cell body and axon (marked in the inset). The site of axotomy using femtosecond laser is labelled using a red arrow. (B) Representative images and schematics of PVD neurons labeled with GFP and mCherry::RAB-3 (*kyIs445;wdIs52*) in green, magenta and merge channels at 3, 24, and 48h post-axotomy (red arrow) at L4 stage. Following axotomy, the axon shows regenerative growth from the severed end (green traces) with mCherry::RAB-3 on the tip (yellow arrowhead), converted neurites (blue traces) which are adjacent dendrites showing mCherry::RAB-3 localization (orange arrowheads), and ectopic neurites from the cell body and proximal axon (orange traces

in schematics). Axonal injury is marked using red arrow. (C) Quantification of the axon regeneration at 3, 24, and 48h following axotomy in the form of growth from the severed end, length of the converted neurites, and ectopic neurites. N = 3–5 independent replicates, n (number of regrowth events) = 17–26. (D) The confocal images of axon regrowth events at 24h post-axotomy in the wildtype, *dlk-1(0)*, *mlk-1(0)*, and *dlk-1(0);mlk-1(0)* mutants with their representative images and schematics. The schematics show regenerative growth from the severed end (green traces) and ectopic neurites as orange traces. (E-F) Axon regeneration is quantified as growth from the severed end (E), and ectopic neurites (F). N = 5–7 independent replicates, n (number of regrowth events) = 17–26. Statistics, For C,E & F, *P<0.05, **P<0.01, ***P <0.001; ANOVA with Tukey's multiple comparison test. Error bars represent SD. ns, not significant.

## Dendrite regeneration in PVD neurons is independent of conventional axon regeneration pathways

Axon regeneration is also controlled by pathways other than the DLK-1 pathway [42]. We tested some of the major genetic regulators implicated in axon regrowth. Axon regeneration is controlled by conserved Calcium and cAMP cascade in many organisms [12,43]. After an axonal injury, there is a calcium influx [11,12], which triggers a cAMP cascade near the injury site and activates DLK-1 MAP3K [15]. An elevation of either intracellular calcium using a gain of function mutation in L-type voltage-gated calcium channel *egl-19* or an elevation of cAMP due to the loss of neuronal phosphodiesterase *pde-4* promotes axon regeneration [12]. However, we observed that neither *egl-19(gf)* nor *pde-4(0)* influenced any aspect of dendrite regeneration (Fig 4A, 4B and 4C). After 24h of dendrotomy, the dendrite was able to regenerate to a similar extent as wild type and the reconnection events were also similar to the wild-type (Fig 4A, 4B and 4C).

The *let-7* miRNA and its downstream target *lin-41* regulate axon regeneration pathway and fusion phenomena [44–46]. Loss of function mutants of *let-7* and *lin-41* showed dendrite regrowth and reconnection comparable to that of the wild type at 24h post-dendrotomy (Fig 4A, 4B and 4C).

PTEN/AKT pathway was previously implicated to play an important role in the regeneration of both axons and dendrites [47,48]. The territory length and reconnection events at 24h post-dendrotomy were not affected in *akt-1* mutant (Fig 4A, 4B and 4C), suggesting dendrite regeneration in PVD neurons is independent of *akt-1*.

The Phosphatidylserine (PS) exposure pathway has emerged as a critical injury sensing mechanism during axonal injury and dendrite degeneration [49,50]. Upon injury, the PS signal activates axon regeneration mechanisms such as DLK/MLK p38 MAPK pathway [51] or fusogen related repair pathway [52]. The PS signal involves exposure of PS to the outer leaflet membrane of the injured neuron through the ABC transporter, CED-7, and further activation of the downstream effectors such as CED-2/CED-5/CED-12 GEF complex and CED-10 GTPase. This signal subsequently activates p38 cascade involving MLK-1 [51]. We did not see any effect in dendrite regeneration parameters in *ced-7*, *psr-1* and *ced-12* mutants (Fig 4A, 4B and 4C). However, loss of *ced-10* showed a drastic impact on dendrite regeneration, including regrowth and fusion phenomena (Fig 4A, 4B and 4C). In *ced-10* mutant, a large gap is seen at 24h post-dendrotomy since the regrowing branches fail to reach the distal end of the primary dendrites (red dotted line, Fig 4A). The fusion events were also drastically reduced (Fig 4A and 4C). This indicated that the CED-10/RAC GTPase might have a novel role in the injury response.

The characterization of different axon regeneration pathways in our dendrite regeneration assay indicated that most known effectors of axon regeneration are not required for dendrite regeneration in PVD neurons. Nevertheless, the substantial reduction of dendrite regeneration in *ced-10* mutant raised exciting questions to explore further.

## CED-10 RAC GTPase is required in neuron for dendrite regeneration

Among the candidate genes tested in our dendrite regeneration assay, the *ced-10* mutant showed a strong reduction in dendrite regeneration (Fig 4A, 4B and 4C). Therefore, we

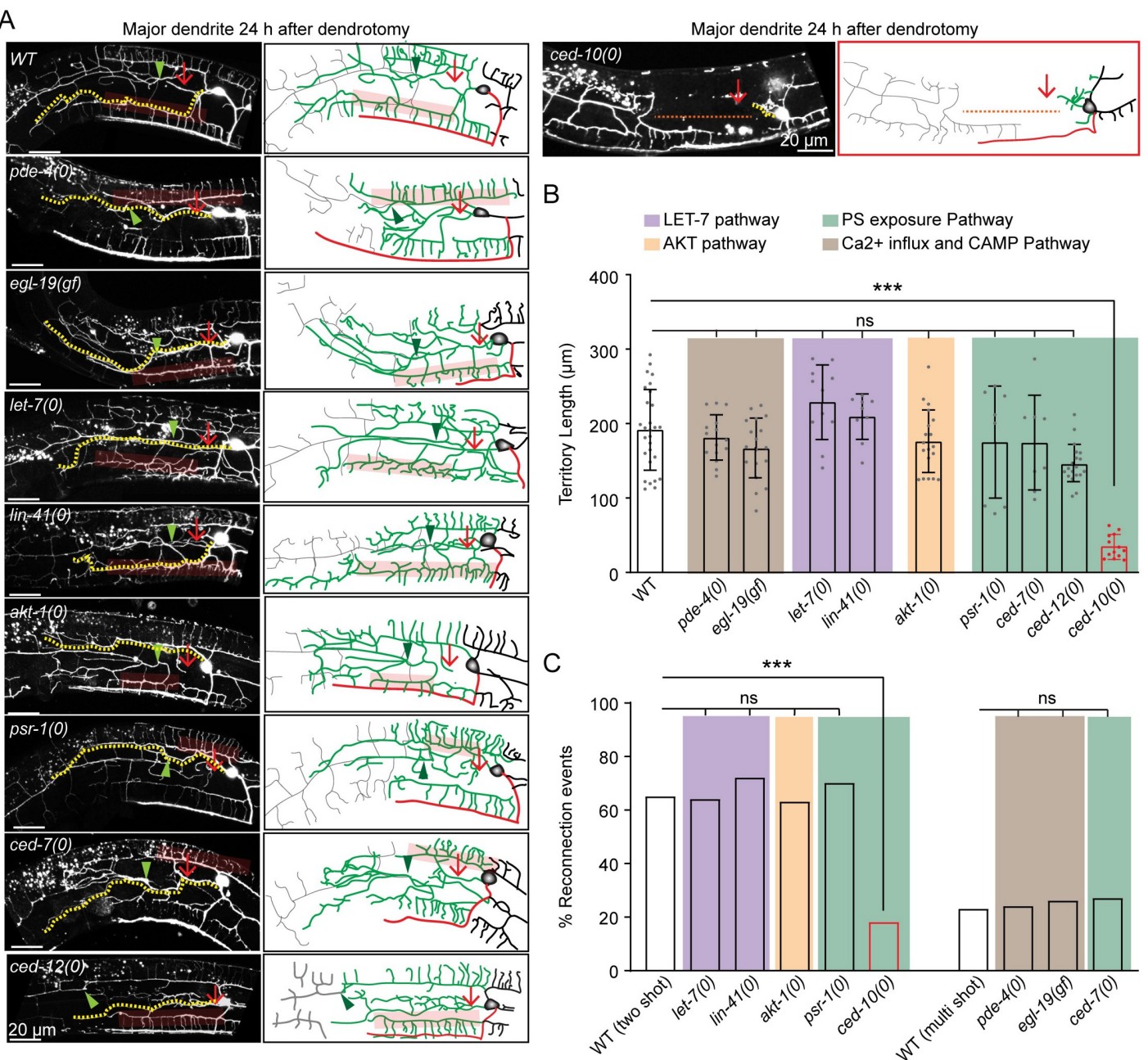

**Fig 4. Dendrite regeneration is independent of conventional axon regeneration molecules.** (A) Confocal images of the major dendrite regeneration events in the wildtype, *pde-4(0)*, *egl-19(gf)*, *let-7(0)*, *lin-41(0)*, *akt-1(0)*, *psr-1(0)*, *ced-7(0)*,*ced-12(0)*, and *ced-10(0)* backgrounds. This experiment is conducted in a strain expressing the *wdIs52* (*pF49H12.4*::GFP) reporter. The illustrations on the right indicating site of dendrotomy (red arrow), regenerated dendrites (green), distal dendritic part (grey), territory length (yellow dotted lines), reconnection events (green arrowheads) and menorah-menorah fusion (faded red boxes). (B-C) The quantification of the territory length (B) and the percentage of reconnection events (C) done from the dendrotomy experiments shown in A. For B & C, N = 3–4 independent replicates, n (number of regrowth events) = 8–19. Statistics, For B, One-way ANOVA with Tukey's multiple comparison test considering $p < 0.05^*$, $0.01^{**}$, $0.001^{***}$, and for C, Fisher's exact test considering $p < 0.05^*$, $0.01^{**}$, $0.001^{***}$. Error bars represent SD. ns, not significant.

investigated the requirement of CED-10 in dendrite injury response in detail. CED-10 is a RAC GTPase involved in regulating cytoskeleton in various morphogenesis processes [53,54]. The RAC family GTPases have been implicated in the growth cone navigation during axon

development [55]. Therefore, we checked whether *ced-10* mutant causes any developmental phenotype in PVD neuron. The length of the axon in PVD neuron remained unaffected in *ced-10* mutant (S3A and S3B Fig). Dendrites also seemed normal in *ced-10(0)* (S3A Fig). The *ced-10* mutant did not affect the axon regrowth parameters in PVD neuron (S3C, S3D and S3E Fig). To understand the requirement of CED-10 in the initiation of dendrite regeneration, we checked at early time points after dendrotomy (Fig 5A). Both the number of filopodia-like structures (red arrowheads, Fig 5A) and the territory length showed a significant reduction in *ced-10(0)* as compared to the wild type at 6h post-dendrotomy (Fig 5A, 5C and 5D). Conversely, when an activated form of CED-10 (G12V) is expressed in the PVD in wild type background, we found that the number of regrowing branches from proximal dendrite increased at 6h post-dendrotomy (Fig 5A, 5C and 5D). Since the higher concentration transgenic lines (10ng/ul) led to the formation of ectopic branches around the cell body region without even performing dendrotomy, we selected a low concentration line (5ng/ul) with a significantly milder developmental defect for our dendrotomy experiment (S3H and S3I Fig). Similarly, the territory coverage length was also increased due to CED-10 activation (Fig 5C). Another gene that codes for RAC GTPase is *mig-2*, which collaborates with CED-10 during development [53,56] (S3A Fig). Although the loss of *mig-2* affected the development of PVD axon (S3A and S3B Fig), the primary major dendrite regrowth was unaffected in this mutant (S3F and S3G Fig). This indicated that developmental impairment of axons would not necessarily affect the dendrite regeneration process. This also indicated a specific requirement of CED-10 in the dendrite regeneration of PVD neurons as axon regeneration was unaffected in the loss of function of *ced-10* (S3C, S3D and S3E Fig).

To check the tissue-specific requirement *ced-10* gene in dendrite regeneration, we expressed the wild type copy of *ced-10* under various promoters. We found that when *ced-10* was expressed under pan-neuronal (*prgef)* or PVD-specific promoter *pser2prom3*, the territory length, branch number, % menorah-menorah fusion and reconnection events were completely rescued in *ced-10* mutant background (Fig 5B–5F). Surprisingly, when *ced-10* was expressed under the epidermal promoter, *pdpy-7* and seam cell promoter, *pgrd-10*, we saw a significant rescue of both the reconnection and menorah-menorah fusion, although the territory length and branching were not rescued in this background (Fig 5C–5F). This suggests that CED-10 RAC GTPase is working cell-autonomously for dendrite regeneration but may also facilitate dendrite regeneration cell-non autonomously by working in nearby epidermal cells.

## TIAM-1 GEF acts upstream of CED-10 in dendrite regeneration

To understand the molecular mechanism by which CED-10 GTPase controls dendrite regeneration in PVD neuron, we speculated that CED-10 could be activated by the upstream factors after dendrotomy. The RAC GTPases get activated upon the removal of the GDP from their GTP binding domain. This is facilitated by the enzymatic activity of Guanine Exchange Factors (GEFs) [57]. There are some known GEFs for CED-10 such as UNC-73 (Trio), TIAM-1 (RhoGEF) and CED-12 (ELMO1), which contain the RAC binding sites, DH (Dbl homology) —PH (Pleckstrin homology) domains [57,58]. To identify the relevant GEF of CED-10 in dendrite regeneration, we have performed dendrotomy in the mutants for these GEFs. Although the axons are predominantly missing in the *unc-73(0)* (S4A Fig), the dendrite regeneration was unaffected (Fig 6). Similarly, the *ced-12* mutant did not affect any parameters of dendrite regeneration (Fig 6). However, the territory length, regenerative branching, and reconnection events were significantly reduced in the absence of *tiam-1* (Fig 6). The phenotype was very similar to what was seen in the *ced-10* mutant. These phenotypes were completely rescued when the wildtype copy of *tiam-1* was expressed in the PVD neuron under *pser2prom3* (PVD

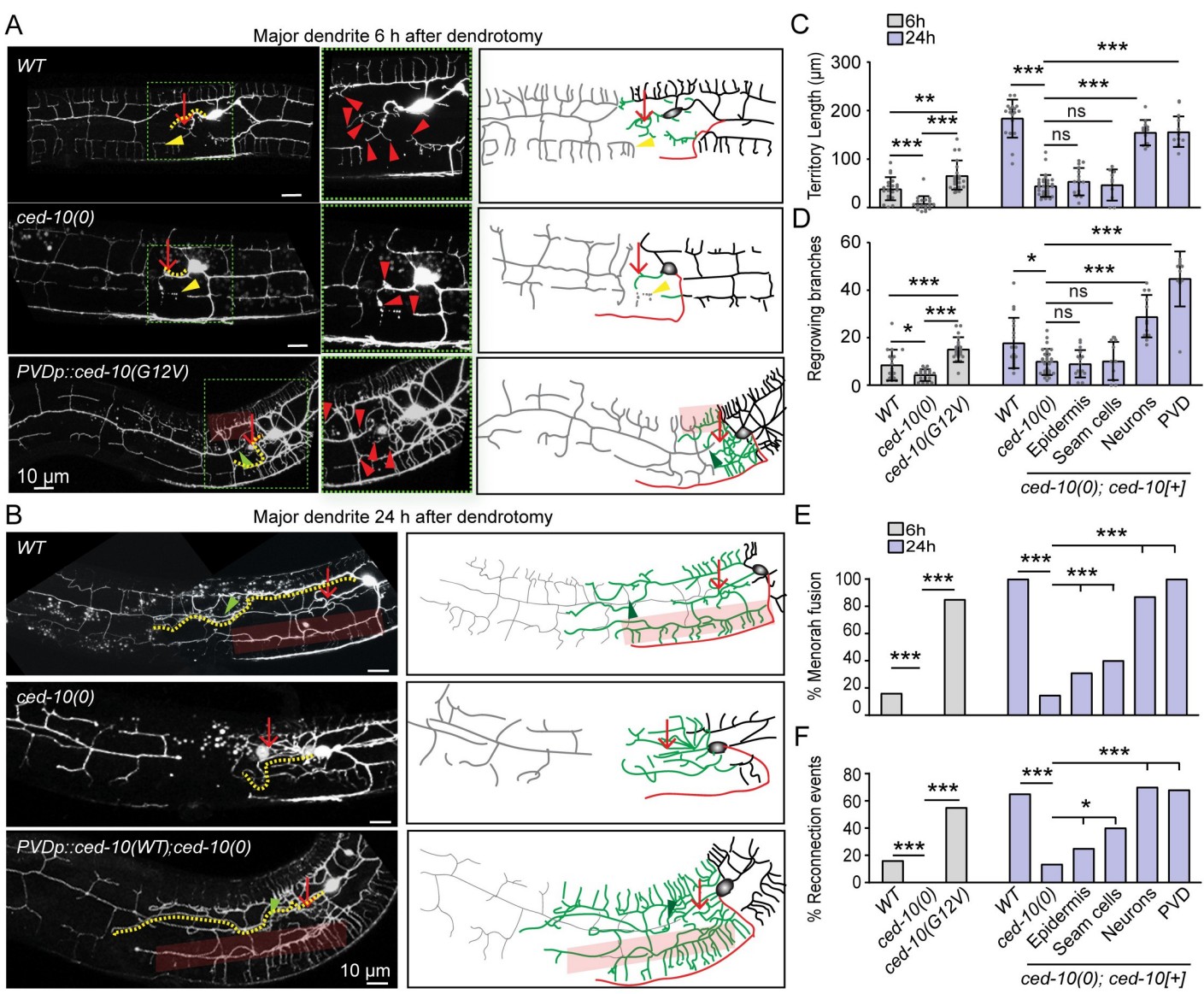

**Fig 5. CED-10 is required in neuron for dendrite regeneration.** (A) Confocal images of PVD in the wildtype, *ced-10(0)* and *pser2prom3::CED-10(G12V)* (Activated CED-10) at 6h post-dendrotomy along with their schematics indicating site of dendrotomy (red arrow), regrowing dendrites (green), distal part (grey), territory length (yellow dotted lines), reconnection phenomena (green arrowheads) and menorah-menorah fusion (faded red boxes). Red arrowheads represent the filopodia like structure at the tip of the cut dendrite, yellow arrowheads represent degenerating menorah. (B) Confocal images of PVD in the wildtype, *ced-10(0)* and *pser2prom3::ced-10(WT); ced-10(0)* at 24h post-dendrotomy along with their schematics. (C-F) The quantification of territory length (C), number of regrowing branches (D), the percentage of reconnection events (E), and percentage of menorah-menorah fusion events (F) in the wild type, *ced-10(0) and pser2prom3::ced-10(G12V)* at 6h post-dendrotomy. For the 24h post-dendrotomy time point, the data from the wild type, *ced-10(0)*, *pdpy-7::ced-10(WT)*(epidermis);*ced-10(0)*, *pgrd-10::ced-10(WT)*(seam cells);*ced-10(0)*, *prgef-1::ced-10(WT)*(All neurons);*ced-10(0)* and *pser2prom3::ced-10(WT)*(PVD);*ced-10(0)* genetic backgrounds were presented. For (C-F), N = 3–5 independent replicates, n (number of regrowth events) = 10–26. Statistics, For C-D, One-way ANOVA Tukey's multiple comparison test considering $p < 0.05^*$, $0.01^{**}$, $0.001^{***}$. For E-F, Fisher's exact test taking $p < 0.05^*$, $0.01^{**}$, $0.001^{***}$. Error bars represent SD. ns, not significant.

specific) promoter (Fig 6). On the other hand, axon regeneration was not affected in the absence of *tiam-1* (S4C and S4D Fig). Since the quarternary branches are missing in the *tiam-1* mutant, it is possible that the lack of dendrite regrowth in this mutant could be a consequence of its developmental defect. Therefore, we tested the dendrite regeneration in *mec-3(0)* mutant,

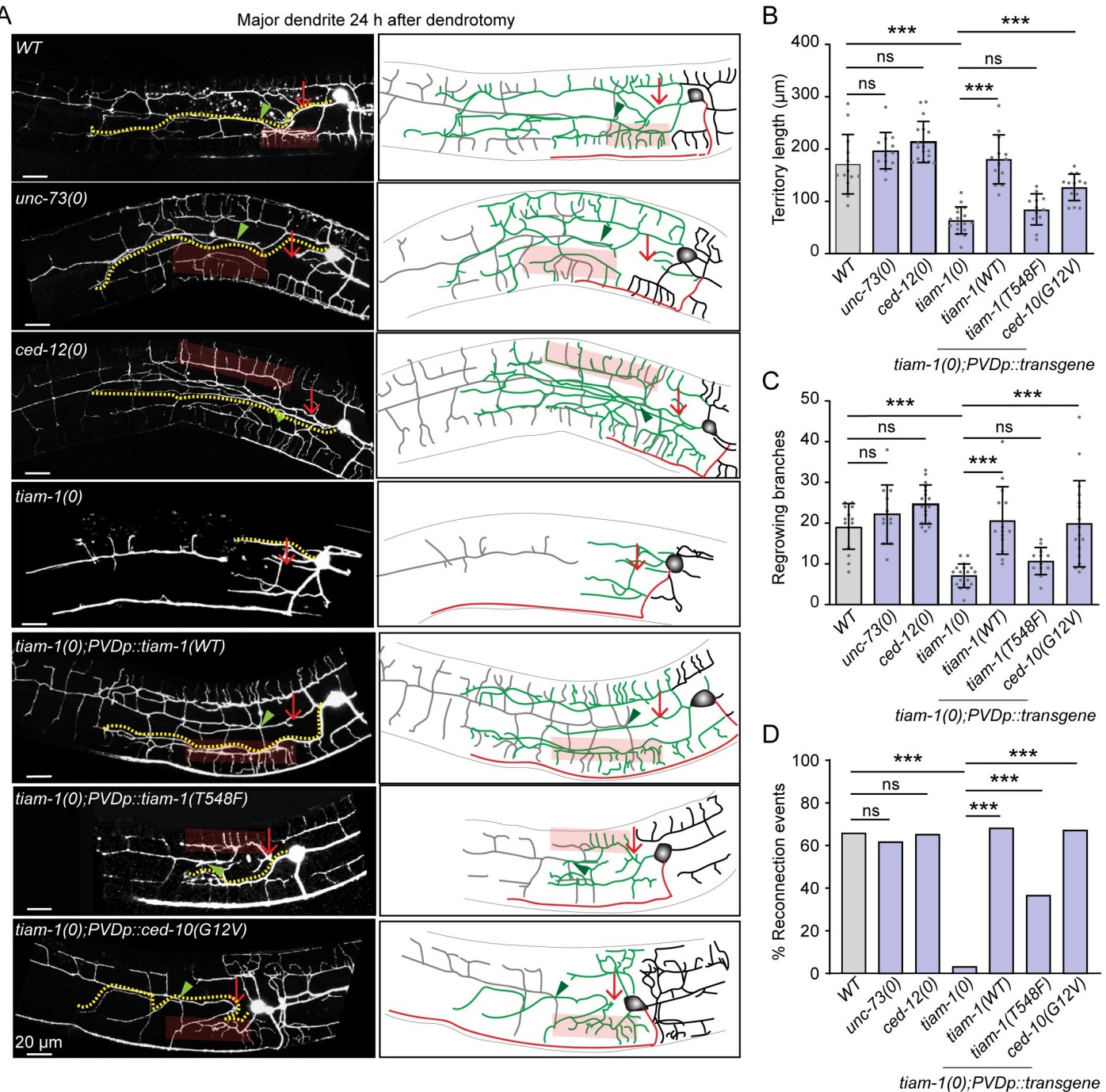

**Fig 6. TIAM-1 acts upstream to CED-10 for dendrite regeneration.** (A) Confocal images of dendrotomized PVD neuron in the wild type, *unc-73(0)*, *ced-12(0)*, *tiam-1 (0)*, *pser2prom3::tiam-1WT (5ng); tiam-1(0)*, *pser2prom3::tiam-1(T548F)5ng;tiam-1(0)* and *pser2prom3::ced-10(G12V) 5ng;tiam-1(0)* at 24h post-injury. The regrowing dendrites, axon, and the distal part of dendrite are represented in green, red, and grey color, respectively. The green arrowheads represent reconnection events and the faded red rectangular boxes indicate menorah-menorah fusion event. (B-C) The quantification of territory length (B), and the the number of regrowing branches (C) in the above mentioned genotypes. N = 3–5 independent replicates, n (number of regrowth events) = 10–29. (D) The bar chart represents the percentage of reconnection events for the above mentioned genotypes, N = 3–5 independent replicates, n (number of regrowth events) = 10–29. Statistics, For B-C, one-way ANOVA with Tukey's multiple comparison test, p<0.05*, 0.01**, 0.001***. For D, Fisher's exact test, p<0.05*, 0.01**, 0.001***. Error bars represent SD. ns, not significant.

in which the higher-order branches were completely absent (S4A and S4B Fig) as reported before [59]. Upon dendrotomy in *mec-3(0)*, the primary dendrites regrew and the territory length was comparable to that of the control (S4E, S4F and S4G Fig). The proximal dendrites in *mec-3(0)* also could reconnect with their distal counterparts (S4E–S4H Fig). Therefore, a defect in dendrite regeneration may not always be correlated to the lack of developmental branching in PVD neuron. Moreover, we addressed whether GEF activity of TIAM-1 would be critical for dendrite regeneration. Since the GEF activity of TIAM-1 was not required for developmental branching of PVD neuron, the expression of a GEF dead mutant of TIAM-1 (T548F) rescued the developmental branching phenotype in *tiam-1* mutant (S4A and S4B Fig) as seen before [60]. However, upon performing dendrotomy in *tiam-1(0)* expressing *pser-2prom3-tiam-1*(T548F), we found that the territory covered by the PVD dendrites and number of branches were not rescued in this background (Fig 6). Therefore, GEF activity of TIAM-1 is specifically required for dendrite regeneration.

To test whether CED-10 activation is limiting in *tiam-1* mutant background, we expressed the constitutively activated form of *ced-10* in PVD neuron in the *tiam-1(0)*. The activated form of CED-10 could bypass the requirement of TIAM-1 in both territory extent as well as reconnection phenomena in dendrite regeneration (Fig 6B and 6C). Thus, the RhoGEF TIAM-1 acts upstream of CED-10 GTPase for dendritic regeneration.

## Discussion

In this report, we presented a detailed analysis of the dendrite and axon regeneration in PVD neurons. Using this system, we could compare the roles of axon regeneration machinery in both dendrite and axon regeneration in the same neuron. Our study revealed a novel function of CED-10 RAC GTPase and TIAM-1 GEF in dendrite regeneration. TIAM-1/CED-10 cascade is required cell-autonomously in PVD to initiate dendrite regrowth and subsequently for branching. Additionally, CED-10 is required in the epidermal cell for regenerative self-fusion events in the same neuron (Fig 7). This expanded our understanding of the mechanism of dendrite regeneration.

### PVD neuron as dendrite regeneration model

Dendrite regeneration is poorly studied as compared to axon regeneration. Few recent studies using the *da* neurons in *Drosophila* have shed some light on the mechanism of dendrite regeneration following laser-assisted surgery [18,19,21]. Both intracellular as well extracellular machinery control the dendrite regrowth in the *da* neurons [18,20,22,61]. However, the signaling mechanism and downstream effectors that lead to dendrite regeneration is unclear. The dendrites of PVD neuron in *C. elegans* have a stereotypic and an elaborate structure [62]. Laser surgery on PVD dendrites leads to self-fusion between the proximal and distal dendrites [30]. The fusion events during dendrite regeneration are driven by the fusogenic activity of AFF-1 [31]. Experiments described in this work allowed addressing the mechanism of dendrite regrowth and branching along with the fusion process. A comprehensive analysis of the requirement of axon regeneration pathways in dendrite regeneration clearly indicated that the regeneration response to dendrotomy in PVD neuron is largely independent of axon injury response pathways including DLK-1. This is consistent with the finding using the *da* neurons in fly [21]. Our findings showed that the dendrite regeneration in PVD neuron involves regrowth, branching, and fusion events between the distal and proximal primary dendrites. This is also seen in the axon of mechanosensory PLM neurons where axotomy leads to both regrowth and fusion phenomenon [12,63]. The finding that the *ced-10* mutant affects both regrowth and fusion events indicated its role in the early response to dendrotomy. Loss of

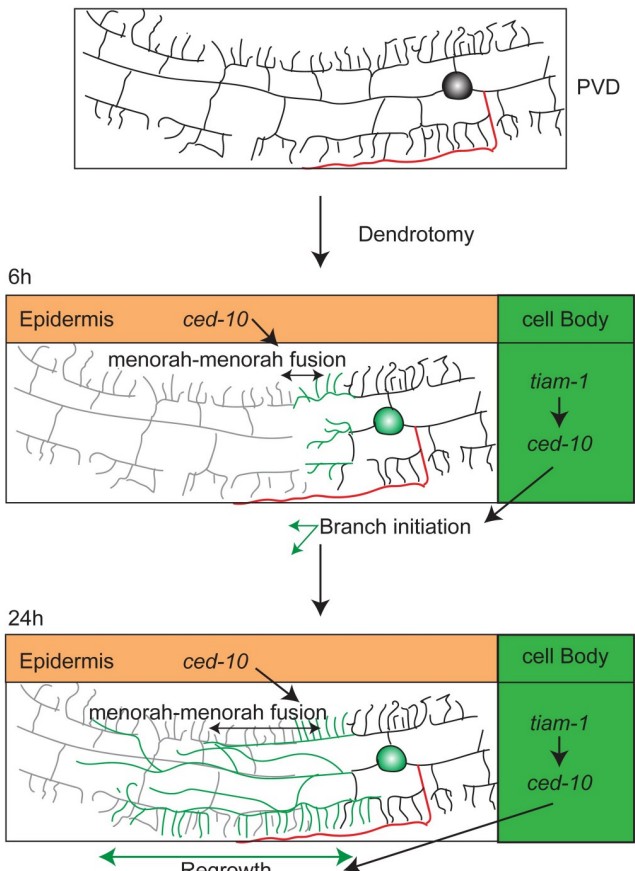

**Fig 7. A model describing the roles of CED-10 and TIAM-1 in dendrite regeneration process in PVD neuron.**

*tiam-1*, the GTP exchange factor for CED-10 GTPase also showed similar perturbation in regrowth and fusion events following dendrotomy. These results indicated that this conserved RAC GTPase and its upstream activator TIAM-1 orchestrate the early events after dendrite injury.

PVD neuron is known for its role in harsh touch response and proprioception of *C. elegans* [29]. In future, the anatomical features of dendrite and axon regeneration can be correlated with behavioral assays using the PVD neuron as an injury model.

## Role of RAC GTPase in dendrite regeneration

The RAC GTPases have been known to play a broader role in developmental processes involving cell debris engulfment [64,65], migration, and axon guidance [53]. The RAC GTPases control these developmental processes through various downstream effectors of actin and microtubule cytoskeleton [55,66,67]. The previous finding that *ced-10* mutant had a partial reduction in the axon regrowth in *C. elegans* motor neuron [51] prompted us to test its role in dendrite regeneration. It was seen that *ced-10* acts genetically downstream to Phosphatidylserine (PS) signal to activate DLK/MLK MAP Kinase pathway in regeneration [51]. However, we did not observe any effect on dendrite regeneration parameters in the mutants affecting either PS or in *dlk-1/mlk-1* mutants. The cell-autonomous role of TIAM-1 and CED-10 in dendrite

regeneration is novel. The RAC GTPases are well-known for their role in F-actin dynamics [68,69], and actin dynamics is a major player in dendritic remodelling during neuronal plasticity [70,71]. It might be possible that CED-10 GTPase induces optimal F-actin dynamics suited for regrowth and branching that is observed after dendrite injury.

Our finding that CED-10 plays a cell non-autonomous function in the surrounding epithelial cells for the menorah-menorah fusion events during regeneration is very intriguing. It was seen that for the fusion to take place, AFF-1 fusogen was delivered from the surrounding seam cells, which are of epidermal origin. CED-10 may initiate the epidermal response to dendrotomy, which might lead to the release of vesicles containing AFF-1 from epidermal cells. Epidermal cells are known for responding to dendrite injury. In case of *da* neurons in *Drosophila*, the PS pathway in epidermal cells controls the dendrotomy induced engulfment of degenerated distal dendrites [50,72]. Therefore, in case of PVD neuron as well, the surrounding epidermis might regulate the fate of the injured dendrites.

## Materials and methods

### *C. elegans* strains and genetics

The *C. elegans* strains were grown and maintained at 20˚C on OP50 bacterial lawns seeded onto Nematode Growth Medium (NGM) plates [73]. The loss of function mutation is represented as (0) and gain of function alleles are represented as (*gf*), for example, the loss of function allele of *dlk-1*, *tm4024* is represented as *dlk-1(0)* and gain of function allele of *egl-19*, *ad695* is represented as *egl-19(gf)*. The mutants used in this study are mostly deletion or substitution mutants unless otherwise mentioned (S1 Table). These mutants were taken from Caenorhabditis Genetics Centre (CGC) and genotyped using their respective genotyping primers.

### Molecular cloning and creating transgenes

Destination vector with PVD neuron-specific promoter, *pser2prom3* [4.1kb]::Gateway [pNBRGWY99] was made by InFusion cloning (Takara). *ser2prom3* Promoter region was amplified from the fosmid WRM0623bG06 using the primers: 5'-ccatgattacgccaagtaaaagtttagtaaattaactgc-3' and 5'-tggccaatcccggggtatgtgttgtgatgtcac-3' and GWY vector backbone was amplified from pCZGY553 using 5'-ccccgggattggcca-3' and 5'-ttggcgtaatcatgg-3'.

For pan-neuronal and epidermal rescue of *ced-10*, *prgef*::GWY [pCZGY66] and *pdpy-7*::GWY [pNBRGWY44], respectively, were recombined with the entry clone of *ced-10 WT* [pNBRGWY88] [74] using the LR recombination (Invitrogen).

For PVD specific expression of wildtype and constitutively active *ced-10*, pNBRGWY99 was recombined with *ced-10 WT* [pNBRGWY88] and *ced-10 constitutively active (G12V)* [pNBRGWY89] entry clone plasmids [74], respectively, using the LR recombination (Invitrogen).

For PVD specific expression of *tiam-1*, *tiam-1* cDNA was ligated with the *pser2prom3* containing backbone DNA from pNBRGWY99 by InFusion cloning. Primers used to amplify tiam-1 cDNA from total cDNA were 5'-tccgaattcgcccttatgggctcacgcctctca-3', and 5'-aaggaacatcgaaattcaaaatagcagctttcttgtaca-3'. The primers used to amplify vector backbone are 5'-cagctttcttgtaca-3' and 5'-aagggcgaattcgga-3'.

The GEF dead *tiam-1* (T548F) [60] substitution mutant was generated using Q5 Site-Directed Muatgenesis kit using the primers 5'- attgttggtcTTTgagaagaaatatgtcagcgatc-3' and 5'-tcttgcagagccatcgcc-3' in *pserprom3*::*tiam-1* (wild-type cDNA, NBR46).

The clones made using various molecular techniques were then injected in the gonads of young adult worms along with co-injection markers such as *pttx-3*::RFP or *pmyo-2*::*mCherry* and the F1 progeny were then isolated and checked for the formation of transgenic lines. Each

transgene was checked for two high transmission lines. List of transgenes used in this study are listed in S2 Table.

## Laser system, dendrotomy and axotomy

Dendrotomy and axotomy experiments were conducted at the L4 stage of *C. elegans* using the Bruker ULTIMA multi-photon microscopy system coupled with two SpectraPhysics tunable Infra-red femtosecond lasers ($\lambda$ = 690–1040 nm) having automated dispersion compensation (MaiTai with DeepSee)[45]. The output of lasers was controlled using Conoptics pockel cells with superior temporal resolution (~$\mu$s) and simultaneous imaging and severing experiments was performed by two independent set of X-Y scan galvanometer mirrors. All the multi-photon images reported here (unless otherwise mentioned) were done using 920 nm (pulse width = 80 fps) and laser ablation was done at 720 nm laser (pulse width = 80 fps, irradiation time: 20ms, avg. laser power: 23mW) using a water immersion Olympus objective (60X / 0.9 NA). Worms were immobilized using either Levamisole hydrochloride (10mM) (Sigma L0380000) as a paralyzing drug agent or with polystyrene beads (Polysciences 00876–15) of diameter 0.1 $\mu$m as friction enhancing agent on 5% agarose pads mounted with Corning cover glass (cat No. 2855–25). Nikon Ti-2 microscope equipped with ultraviolet laser (Micropoint laser, 337nm, 3ns Pulse, Max Pulse enery-200$\mu$J, Max Average Power- 4mW) system was also used to assist the laser injury using using 100X/1.40NA oil immersion objective.

PVD dendrites were severed with the first laser shot at the first branch point (~10$\mu$m away from cell body) followed by one (two-shot experiment, Fig 1A) or more (multi-shot experiment, S1A Fig) consecutive shots with a relative distance of 10–15$\mu$m from the previous shot creating a visible gap with no fragments left at the cut site. The axons were severed distally at the segment fasciculating with the ventral nerve cord (VNC) (~50 $\mu$m away from the cell body). A clean transection was ensured by making two shots 5 $\mu$m apart at a distance of 10 $\mu$m away from the distal-most lateral to ventral transition (LV) point of PVDR and PVDL. After severing, worms were recovered from the agarose pad using an aspirator with 1X M9 solution onto freshly seeded NGM plates for further observation.

## Imaging

To observe the regenerative response, injured worms were imaged at 3, 24 and 48 hours (h) after injury. The worms were paralyzed and mounted in 10 mM Levamisole hydrochloride (Sigma) solution on slides containing medium on 5% agarose (Sigma) pads. The worms were imaged with 63X/1.4NA oil objective of Nikon A1R confocal system at a voxel resolution of 0.41 $\mu$m x 0.41 $\mu$m x 1 $\mu$m and tile imaging module using imaging lasers 488 nm(GFP), 543 nm(mCherry/RFP) with 1–1.8 AU pinhole at 512x512 pixel frame size for further analysis.

## Dendrite regeneration analysis and quantification

Dendrite regeneration was quantified based on the regrowth and fusion parameters. The territory covered by regenerated dendrite (Fig 1B, yellow dotted line) was estimated using Simple Neurite Tracer plugin in Fiji-ImageJ by tracking the longest regrowing dendrite from the cell body to the tip of the dendrite using hessian based tracking onto the Z stack of PVD neuron. 'Total number of branches' were then also calculated using the same plugin from the point of emergence (cell body or dendrite) to the termination of every branch.

PVD dendrites have the capability to fuse after injury [30,31]. We also observed reconnection between the regenerating proximal primary dendrite with distal primary dendrite or distal menorah (Fig 1B, green arrow), which were evaluated as reconnection events using 'depth coded' confocal images, the population of worms showing proximal and distal dendrites on

the same depth was calculated and represented as a fraction showing reconnections. Long menorah having more than one secondary dendrite connected to it (Fig 1B, faded red rectangular boxes), was classified as menorah-menorah fusion, and the percentage of PVD neurons showing menorah-menorah fusion was quantified. Following injury, distal dendrite shows hallmarks of Wallerian degeneration [75,76], which were estimated as the percentage of PVD neurons showing distal part degeneration (Fig 1C, grey dotted neurites).

## Axon regeneration analysis and quantification

Analysis of the axon regeneration was carried out by ImageJ. Neurites were traced, and their lengths were quantified using the Simple Neurite Tracer plugin of ImageJ. Volumetric visualization using the '3D project' of confocal stacks facilitated a qualitative estimate and descriptors of regenerated neurites based on the mCHERRY::RAB-3 localization. Regenerative growth was classified into the neurite growth from the cut point of the axon, conversion of the adjacent tertiary dendrites to axon like identity as per the localization of mCHERRY::RAB-3 punctae, and ectopic branches from the cell body, proximal axon, or converted neurites.

Quants obtained from ImageJ for both dendritic and axonal regeneration were further analyzed by Excel and Graphpad to get statistical information.

## Statistical analysis

Statistical analyses were compassed using Graphpad Prism software (Prism 8 V8.2.1(441). Two samples were analyzed using the unpaired two-tailed t test. The statistical analysis of multiple samples was performed using One way-ANOVA with Tukey's multiple comparisons test. The data that was used for ANOVA analysis was naturally occurring data having normal distribution spread which was not further processed. To compare the population data, the fraction values with respect to each sample were calculated and compared using the two-tailed Chi-square Fisher's exact contingency test. For each bar plots, the number of samples (n) and the number of biological replicates (N) are mentioned In the respective figure legends. The significance considered for all statistical experiments are $p < 0.05^*$, $0.01^{**}$, $0.001^{***}$.

## Supporting information

**S1 Fig. Regeneration response of primary dendrites of PVD, Related to Fig 1.** (A) The confocal images and illustrations (right) of regeneration phenomena of primary major dendrites of PVD at 3h, 24h and 48h post-dendrotomy using four laser shots. The experiment was performed in worms expressing *wdIs52* (*pF49H12.4::GFP*) reporter. The large gap created at 3h post-dendrotomy due to multiple laser shots is indicated with orange dotted line with double-arrowheads (topmost panel). The faded red box and green arrowhead highlight the menorah-menorah fusion event and reconnection phenomenon, respectively. In the illustration, the regrowing dendrites, the remnants of distal part, and the axon is indicated in green, grey, and red colors, respectively. (B-C) The percentage occurrence of reconnection (B) and menorah-menorah fusion events (C) at 3h, 24h and 48 h after single or multi-shot laser-dendrotomy is represented. N = 3–5 independent replicates, n (number of regrowth events) = 20–60. (D-E) Quantification of territory length (D), N = 3–5 independent replicates, n (number of regrowth events) = 20–70 (D), and the total number of branches (E), N = 3–5 independent replicates, n (number of regrowth events) = 14–34 at 3h and 24h post-dendrotomy using single and multi-shots. (F) Confocal images with schematics showing the regeneration events at 3, 24, and 48h following the injury on primary minor dendrites. (G) Depth coded images of dendrite regeneration events in the wildtype worms along with magnified version of the reconnection area (right) showing the proximal part contacting the distal part or crossing over the distal part.

(H) Quantification of reconnection and menorah-menorah fusion events counted from the depth-coded vs regular z-projected images. (I) Quantification of degeneration of the distal parts in the 'reconnection' vs 'no reconnection' events. For, H-I, N = 3–4 independent replicates, n (number of regrowth events) = 20–30. Statistics, for B-C, Fisher's exact test, taking $p<0.05^*$, $0.001^{***}$. For D-E, one-way ANOVA with Tukey's multiple comparisons method, taking $p<0.001^{***}$. For H-I, unpaired t test, taking $p<0.05^*$, $0.001^{***}$. Error bars represent SD. ns, not significant.
(TIF)

**S2 Fig. The dendrite regeneration does not require DLK/MLK pathway, Related to Fig 2.**
(A) Confocal images of the regeneration events of minor dendrites in wild-type and *dlk-1(0); mlk-1(0)* backgrounds at 24h post-dendrotomy. The schematics representing site of dendritic injury (red arrow), regenerated dendrite (green), distal part (grey), reconnection phenomena (green arrowheads) and menorah-menorah fusion (semi-transparent red boxes). Territory length is represented as yellow dotted lines in the confocal image. (B-C) The territory length (B) and the number of regrowing branches (C) at 3-6h and 24h post-dendrotomy. N = 3 independent replicates, n (number of regrowth events) = 7–19. (D) The genetic pathway involving *rpm-1*, controlling axon growth termination. (E) Representative confocal images showing the developmental phenotype of PVD in various mutants in *rpm-1* pathway. In the schematics, the axon in shown in red. Please note that in *rpm-1* mutants, an overshooting of axon is noticed. (F) The quantification of axonal length of PVD neurons in the wild-type, *dlk-1(0)*, *mlk-1(0)*, *dlk-1(0); mlk-1(0)*, *rpm-1(ju23)*, *rpm-1(ok364)* and *rpm-1(ok364);dlk-1(tm4024)* mutants at L4 stage, N = 3–4 independent replicates, n (number of PVD imaged) = 8–27. Statistics, for B-C & F one-way ANOVA with Tukey's multiple comparison method taking $p<0.05^*$, $0.01^{**}$, $0.001^{***}$. Error bars represent SD. ns, not significant.
(TIF)

**S3 Fig. CED-10 is required for dendrite regeneration, related to Fig 5.** (A) Confocal images of PVD neuron in the wild-type, *ced-10(0)*, and *mig-2(0)* is shown along with its illustrations (right) indicating the axon in red. (B) The axonal defect at L4 stage is calculated as percentage defect. N = 3 independent replicates, n (number of PVD imaged) = 10–12. (C) Confocal images of axon regeneration events in the wild-type, *ced-10(0)* and *mig-2(0)* at 24h post-axotomy along with their schematics indicating site of axonal injury with red arrow, regrowing axon from severed end in green and ectopic neurites in orange color. (D-E) Quantification of axon regeneration as growth from the severed end (D) and length of ectopic neurites (E) in the wild-type, *ced-10(0)* and *mig-2(0)* at 24h post-axotomy. N = 3–4 independent replicates, n (number of regrowth events) = 14–25. (F) Confocal image of dendrite regeneration in *mig-2 (0)* at 24h post-dendrotomy. The illustration indicating the regrowing dendrites in green color, distal part in grey color, reconnection phenomenon with green arrowheads, menorah-menorah fusion with faint red rectangular boxes. (G) The territory length in the wild-type and *mig-2(0)* at 24h post-dendrotomy, N = 3–4 independent replicates, n (number of regrowth events) = 10–11. (H) The confocal images of PVD neuron at L4 stage expressing *pser2prom3::ced-10(G12V)* extrachromosomal transgenes. The *pser2prom3::ced-10(G12V)* plasmid was injected at *1ng/µl*, *5ng/µl*, and *10ng/µl* concentrations to obtain these lines. (I) Quantification of number of ectopic neurites emerging out of cell body or adjacent dendrites at L4 stage in the wild-type and transgenic background expressing *pser2prom3::ced-10(G12V)* extrachromosomal arrays, N = 3 independent replicates, n (number of PVD imaged) = 11–20. Statistics, for B, Fisher's exact test, for D-E & H, one-way ANOVA with Tukey's multiple comparison method, and for G, unpaired t test, $p<0.05^*$, $0.01^{**}$, $0.001^{***}$. Error bars represent SD. ns, not

significant.
(TIF)

**S4 Fig. Developmental phenotype and regeneration phenomena of PVD neuron in Rho/RAC-GEF mutants, related to Fig 6.** (A) Confocal images of PVD neuron in the wild-type, *unc-73(0)*, *tiam-1(0)*, *pser2prom3::tiam-1(T548F);tiam-1(0)*, *pser2prom3::ced-10(G12V);tiam-1(0)*, and *mec-3(0)* background at L4 stage. (B) The percentage of PVDs showing quaternary branches in the wild-type, *tiam-1(0)*, *pser2prom3::tiam-1(T548F);tiam-1(0)*, *pser2prom3::ced-10(G12V);tiam-1(0)* and *mec-3(0)* backgrounds, N = 3 independent replicates, n (number of PVD imaged) = 12–15. (C) Confocal images of axon regeneration at 24h post-axotomy in the wild-type and *tiam-1(0)* backgrounds. The regenerated axon from the severed end is shown in green color in the illustration. (D) The quantification of axon regrowth from the severed end in the wild-type and *tiam-1(0)*, N = 3 independent replicates, n (number of regrowth events) = 10–12. (E) The confocal images of dendrite regeneration at 24h post-dendrotomy in the wild-type and *mec-3(0)* is shown along with their schematics representing the site of injury (red arrow), regenerated dendrites (green), reconnection events (green arrowhead), and the menorah-menorah fusion event (faint red rectangular box). (F-G) The territory length (F) and the number of regrowing branches (G) in the wild-type and *mec-3(0)* at 24h post-dendrotomy, N = 3 independent replicates, n (number of regrowth events) = 11–14. (H) Percentage of worms showing reconnection phenomena at 24h post-dendrotomy in the wild-type and *mec-3(0)*, N = 3 independent replicates, n (number of regrowth events) = 11–14. Statistics, for, B & H, Fisher's exact test, for D & F-G, unpaired t test considering $p < 0.05^*$, $0.01^{**}$, $0.001^{***}$. Error bars represent SD. ns, not significant.
(TIF)

**S1 Table. List of *C. elegans* strains used in this paper.**
(XLSX)

**S2 Table. List of strains carrying extrachromosomal transgenes used in this paper.**
(XLSX)

## Acknowledgments

We thank Yuji Kohara for cDNAs. We thank National BioResource Project (NBRP), Japan, and Caenorhabditis Genetics Center (CGC) for strains. We thank Sandhya Koushika, Yishi Jin, Andrew Chisholm, Kavita Babu, and Cori Bargmann for the help with strains and plasmids. We thank Erik A. Lundquist for providing reagents to manipulate small GTPases. We thank Bhavani Shankar Sahu for his comments on the manuscript.

## Author Contributions

**Conceptualization:** Harjot Kaur Brar, Swagata Dey, Anindya Ghosh-Roy.

**Data curation:** Harjot Kaur Brar, Swagata Dey, Devashish Pande.

**Formal analysis:** Harjot Kaur Brar, Swagata Dey, Devashish Pande.

**Funding acquisition:** Swagata Dey, Anindya Ghosh-Roy.

**Investigation:** Harjot Kaur Brar, Swagata Dey, Smriti Bhardwaj, Devashish Pande.

**Methodology:** Harjot Kaur Brar, Swagata Dey, Smriti Bhardwaj, Pallavi Singh, Shirshendu Dey.

**Project administration:** Anindya Ghosh-Roy.

**Resources:** Smriti Bhardwaj, Pallavi Singh, Anindya Ghosh-Roy.

**Software:** Shirshendu Dey.

**Supervision:** Anindya Ghosh-Roy.

**Visualization:** Harjot Kaur Brar, Swagata Dey, Smriti Bhardwaj, Devashish Pande.

**Writing – original draft:** Harjot Kaur Brar, Swagata Dey, Anindya Ghosh-Roy.

**Writing – review & editing:** Harjot Kaur Brar, Swagata Dey, Anindya Ghosh-Roy.

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
