## [Decision Letter · Decision Letter 0]

1 Sep 2021

Dear Dr. Ghosh-Roy,

Thank you very much for submitting your Research Article entitled 'Dendrite regeneration in C. elegans is controlled by the RAC GTPase CED-10 and the RhoGEF TIAM-1' to PLOS Genetics.

The manuscript was fully evaluated at the editorial level and by independent peer reviewers. The reviewers appreciated the attention to an important problem, but raised some substantial concerns about the current manuscript. Based on the reviews, we will not be able to accept this version of the manuscript, but we would be willing to review a much-revised version. We cannot, of course, promise publication at that time. 

We believe that both reviewers have clearly stated their concerns regarding the manuscript.  All of the concerns of Reviewer #2 and most of Reviewer #1 are considered minor concerns.  However, Reviewer #1 has three major concerns with which we agree. 

If you decide to revise the manuscript for further consideration at PLOS Genetics, please aim to resubmit within the next 60 days, unless it will take extra time to address the concerns of the reviewers, in which case we would appreciate an expected resubmission date by email to plosgenetics@plos.org.

[LINK]

We are sorry that we cannot be more positive about your manuscript at this stage. Please do not hesitate to contact us if you have any concerns or questions.

Yours sincerely,

Kaveh Ashrafi

Associate Editor

PLOS Genetics

Gregory P. Copenhaver

Editor-in-Chief

PLOS Genetics

Reviewer's Responses to Questions

**Comments to the Authors:**

Reviewer #1: This paper uses powerful genetic and live cell imaging approaches to screen for effectors of dendritic regeneration. The experimental strategy exploits the stereotypical and readily visible dendritic branching pattern of the PVD nociceptive neuron in C. elegans. For these experiments, the PVD neuron is labeled with GFP and a laser is used to sever the primary PVD dendrite. Extensive dendritic regeneration was quantified from images collected a later time points. An extensive list of available mutants derived from studies of axon regeneration were tested for potential roles in dendrite regeneration. Although results are largely negative, these findings are important because they are consistent with the emerging evidence that axonal and dendritic regeneration are likely to rely on distinct mechanisms. A key finding of this work is the discovery that mutations that disable conserved effectors of actin dynamics, CED-10/RAC and TIAM-1/GEF, also impair PVD dendrite regeneration. Interestingly, rescue experiments point to both cell autonomous (PVD) and non-cell autonomous (epidermis) roles. A constitutively active version of ced-10 rescues the tiam-1(0) regeneration defect, a finding consistent with previous work showing that TIAM-1 functions as a GEF (guanine exchange factor) to activate CED-10. The experiments are rigorous, the paper is well-written (see minor revisions), and new findings are significant. Additional experiments are necessary, however, to shore up the proposed role of TIAM-1 in the regeneration mechanism as outlined below.

Major Revisions

1. PVD dendritic branching is drastically reduced in tiam-1 mutants which show limited secondary and tertiary branch outgrowth. It thus seems plausible that the overall reduction in PVD dendritic branching in tiam-1 mutants could also hinder regeneration and that this effect would also be observed for mutants in other genes (e.g., hpo-30, lect-2, act-4, dma-1) that drive PVD branching. The authors need to test at least one additional PVD dendritic branching mutant to rule out this possibility.

2. If TIAM-1 GEF activity is required for activating CED-10-dependent regeneration, then a tiam-1 point mutation that specifically eliminates TIAM-1 GEF activity should impair dendritic regeneration (Demarco et al., 2012). This question is important because a recent paper showed that TIAM-GEF activity is apparently not required for PVD dendritic branching (Tang et al., eLIFE, 2019).

3. The authors report that severed PVD dendrites regrow and ultimately fuse with each other to restore a contiguous dendritic arbor. The evidence of fusion is limited to the observation that the tips of apposing regenerated, GFP-labeled dendrites appear continuous in the light microscope. This observation does not rule out the alternative explanation that the regenerated dendrites are overlapping each other or touching but not actually fused. I’m not requiring an experiment to distinguish between these possibilities since this does not seem be a convention in the field but the authors need to address this caveat at the very least in the manuscript. This question is actually quite significant since the long term goal of this work on regeneration in model organisms is to discover pathways that can restore function to injured circuits.

Minor revisions

1. Minor grammatical and stylistic errors are scattered throughout the text. The use of the article “the” is problematic in several instances (e.g., “triggers elevation in the Cyclic Adenosine Monophosphate (cAMP)…” should be “triggers elevation of Cyclic…”

2. Top of page 9: what does “multivariate process” mean?

3. Top of page 14: what does this mean? “Showed a significant decrease in the same.”

4. Pg 18 suggested rewording: “dendrite regeneration assay indicated that most known effectors of axon regeneration are not required for dendrite regeneration in PVD neurons.”

5. Pg 20 What is the meaning of this sentence?, “The number of filopodia like structures (arrowheads, Figure 5A) and territory covered seemed to have decreased in ced-10(0) as compared to wild type…” Figure 5C shows a significant effect for “territory covered” but Figure 5D shows no significant effect for “regrowing branches.”

6. Pg 20. What does “higher concentration lines” mean and why is this notable?

7. Pg 21, “This infers…” should be “This suggests…”

8. Figure 1 The yellow dotted line is very difficult to see. Needs to be brighter with thicker dots.

9. Figure 3. Images of the mCherry::RAB-3 marker (Panel B at 24h and 48h) are unconvincing given large number of fluorescent puncta that seem to be distributed throughout the field of view?

Reviewer #2: In this manuscript Brar et al. perform a basic characterization of dendrite regeneration in C. elegans PVD neurons, compare dendrite and axon regeneration in the same cell, and identify two genes required for dendrite regeneration. This work provides an important foundation for understanding mechanisms that allow neurons to respond to dendrite injury. Dendrite regeneration (with the exception of fusion, which is specific to C. elegans) has been studied almost exclusively in Drosophila. Having another model system in which to study this process is a substantial advance. As in Drosophila, none of the core axon regeneration machinery regulates dendrite regeneration, and these are compared in the same cell. Moreover, the authors find two regulators of dendrite regeneration. This is a nice foundational story that provides good footing to use C. elegans to investigate dendrite regeneration. There are just a few points that would strengthen the manuscript further:

1. Axon regeneration in PVD looks quite subtle; growth difficult to see in control image in figure 3D. Perhaps a different image might help? It looks like the axon does not regrow to reach its former length; is this the case? If so, it would be good to discuss what this might mean for function.

2. In the manuscript loss of function mutations are designated (0). Are these null alleles? For ced-10 it would be particularly helpful to know if the alleles used are null as no developmental phenotype is observed.

3. It would be very helpful to show whether axon regeneration is affected similarly to dendrite regeneration in ced-10 and tiam-1 mutants. As the story stands, it is difficult to know whether either is required broadly for regenerative growth in this cell type, or specifically for dendrite regeneration.

**Have all data underlying the figures and results presented in the manuscript been provided?**

Reviewer #1: **No: **I don't see supplemental files (e.g. excel spread sheets with data for quantitative analysis) for this paper at your site. But, I have no reason to question the results shown in figures and thus do not need to see the original data.

Reviewer #2: Yes

PLOS authors have the option to publish the peer review history of their article (what does this mean?). If published, this will include your full peer review and any attached files.

Reviewer #1: No

Reviewer #2: No

---

## [Decision Letter · Decision Letter 1]

28 Feb 2022

Dear Dr %Gosh-Roy%,

We are pleased to inform you that your manuscript entitled "Dendrite regeneration in C. elegans is controlled by the RAC GTPase CED-10 and the RhoGEF TIAM-1" has been editorially accepted for publication in PLOS Genetics. Congratulations!

Yours sincerely,

Kaveh Ashrafi

Associate Editor

PLOS Genetics

Gregory P. Copenhaver

Editor-in-Chief

PLOS Genetics

Comments from the reviewers (if applicable):

Reviewer's Responses to Questions

**Comments to the Authors:**

Reviewer #1: The Authors have satisfactorily addressed my critiques

Reviewer #2: The authors have made significant new additions to the manuscript. In particular, the experiments with kinase dead Tiam-1 nicely separate the role in development and regeneration. The ability of the mec-3 mutant animals to regenerate dendrites despite simplified morphology is also a nice addition. Showing the Tiam-1 mutant has normal axon regeneration also strengthens the argument that this pathway is specifically required for dendrite regeneration. This paper will make a very nice addition to the newly emerging field of dendrite regeneration.

**Have all data underlying the figures and results presented in the manuscript been provided?**

Reviewer #1: Yes

Reviewer #2: Yes

PLOS authors have the option to publish the peer review history of their article (what does this mean?). If published, this will include your full peer review and any attached files.

Reviewer #1: No

Reviewer #2: No

**Data Deposition**

http://datadryad.org/submit?journalID=pgenetics&manu=PGENETICS-D-21-01021R1

**Press Queries**

---

## [Editor Report · Acceptance letter]

11 Mar 2022

PGENETICS-D-21-01021R1 

Dendrite regeneration in </i>C. elegans</i> is controlled by the RAC GTPase CED-10 and the RhoGEF TIAM-1 

Dear Dr Ghosh-Roy, 

We are pleased to inform you that your manuscript entitled "Dendrite regeneration in </i>C. elegans</i> is controlled by the RAC GTPase CED-10 and the RhoGEF TIAM-1" has been formally accepted for publication in PLOS Genetics! Your manuscript is now with our production department and you will be notified of the publication date in due course.

With kind regards,

Zsofia Freund

PLOS Genetics

On behalf of:
